# 3D Head Shape Feature Analysis of Zika-Infected Children

**DOI:** 10.3390/v16091406

**Published:** 2024-09-03

**Authors:** Xiangyang Ju, Peter Mossey, Ashraf Ayoub

**Affiliations:** 1Medical Devices Unit, NHS Greater Glasgow and Clyde, Glasgow G3 8SJ, UK; 2School of Dentistry, University of Dundee, Dundee DD1 4HR, UK; p.a.mossey@dundee.ac.uk; 3Dental Hospital School, College of MVLS, University of Glasgow, Glasgow G2 3JZ, UK

**Keywords:** Zika virus, congenital Zika syndrome, 3D image analysis

## Abstract

Congenital Zika syndrome (CZS) has been identified a constellation of congenital anomalies caused by Zika Virus (ZKV) infection during pregnancy. The infection with ZKV could lead to microcephaly of the fetus due to a severe decrease in brain volume and reduced brain growth. The preliminary screening of CZS is based on measuring head circumference; the diagnosis is made if this measurement is below two standard deviations below the mean. The analyses of the 3D head features of infected infants are limited. This study analyzed 3D head images of 35 ZKV-positive cases with an average age of 16.8 ± 2 months and 35 controls with an average age of 14.4 ± 5 months. This study focused on identifying potential diagnostic characteristics of CZS. The 3D head images were captured using a 3D imaging system. The averaged images of the two groups were aligned to illustrate the size and shape differences. There were significant differences in centroid size, head circumference (HC), head height (HH), and chin height (CH) between the two groups. We also identified significant differences in the indices of chin height/total facial height (CH/TFH) and head height/head circumference ratio (HH/HC) between the CZS and control cases. An HH/HC of 0.49 showed a sensitivity of 0.86 and a specificity of 0.74 in diagnosing CZS, which is more sensitive than the routinely used HC measurement. The index of HH/HC has potential to be used as the gold standard for the early screening for the detection of CZS cases.

## 1. Introduction

In February 2016, the World Health Organization (WHO) reported that the epidemic of Zika Virus (ZKV) in the Americas constituted a public health emergency of international concern [1]. The WHO called for support in scaling up and strengthening the surveillance systems in countries that have reported cases of microcephaly and other neurological conditions that may be associated with the ZKV infection.

Moore et al., 2017 [2] reported five features that were rarely seen with any other congenital infections or were unique to congenital Zika virus infection, which include severe microcephaly with partially collapsed skull, thin cerebral cortices with subcortical calcifications, macular scarring and focal pigmentary retinal mottling, congenital contractures, and marked early hypertonia in addition to the symptoms of extrapyramidal involvement. However, some of the dysmorphic characteristics observed in CZS are common to other congenital malformations.

It is recommended that the head circumference should be measured within 24 h of birth to identify congenital microcephaly [3]. Congenital microcephaly was defined as a head circumference (HC) more than two standard deviations below the normal mean as defined in the WHO child growth standards [4]. However, about 20% of infants with a normal HC have developed microcephaly at a later stage [5]. Therefore, the current measuring of HC is relatively unspecific and based on skull phenotype, which is too broad to capture microcephaly of CZS. It is prudent to explore other measurements that would strengthen the sensitivity and specificity of the surveillance of CZS. Ju et al. [6] applied Canonical Correlation Analysis (CCA) on the 3D images of the head of CZS children; they reported a significant correlation in the first CCA variates of face and vault. Ayoub et al. [7] reported that Spearman’s rank–order correlation coefficients show that the head height (HH) has a stronger correlation (0.87) in ZKV-infected infants than the routinely used HC. Fonteles et al. [8] used standardized photographic images and clinical assessments to investigate the facial pattern of infants born with CZS in comparison with clinically healthy children. They reported that seven indices could detect the differences between the CZS and control groups; these included the following: midfacial height/horizontal facial reference, inter-alar distance/horizontal facial reference, nasal root depth/midfacial height, posterior nasal length/midfacial height, vertical position of the ear/midfacial height, ear length/midfacial height, and chin height/total facial height (CH/TFH). The CH/TFH index showed 93.9% sensitivity and 80.6% specificity in diagnosing CZS. However, their investigation was limited to 2D linear measurements of facial features in CZS and did not measure the 3D morphology of the vault of the skull which is key feature of this craniofacial deformity.

In this study, we analyzed 3D head images of 35 Zika cases and 35 controls to extract key distinctive features of the face and vault of the skull. We measured HC, which is routinely used for the surveillance of CZS. We also evaluated key parameters, which include chin height (CH), total facial height (TFH), and head height (HH). In addition, we measured the indices of HH/HC and CH/TFH, which were recommended for the diagnosis of CZS. The main objective of this study was the exploration of more sensitive measurements and facial characteristics to improve the sensitivity of the surveillance for the diagnosis of CZS.

## 2. Materials and Methods

### 2.1. 3D Images of Control and Zika-Infected Children

Ethical approval for this study was obtained from the Brazilian ethics committee (CAAE 60031216.4.0000.5420). Infants of positive Zika-related IgM were recruited. A 3D stereophotogrammetry imaging system (Di3D v6.8.9, Dimensional Image Ltd., Glasgow G52 4RU, UK) was used to capture 3D images of the heads of the children diagnosed with CZS at Robert Santos General Hospital, Salvador, Brazil. A control group of healthy infants, from the same geographic region, were included in the study for comparison. The Di3D imaging system had an accuracy of 0.5 mm in capturing the shapes of the heads. For each infant, a custom-made disposable elastic cap was used to cover the head of the children. The children were held by their mothers to allow multiple images of the face, including the right and left profiles, the vault of the skull, and the back of the head. The 3D models of the craniofacial region were reconstructed to create a 360-degree model of the face and the cranium using specially designed and developed Di3D software (Di3DView v6.8.9) for this purpose. The developed 3D images of the face and the vault of the skull were saved in the Wavefront OBJ file format for analysis. The head models of 35 CZS children and 35 controls were analyzed in this study.

### 2.2. Conformation Process of Faces and Vaults

A generic mathematical mesh was conformed through elastic deformation on the shapes of individual 3D images of the cranial vault and the face. This maintained the topological mathematical configurations of the mesh for analysis. The 3D conformed meshes have the same correspondences of vertices of the captured images of the face and the cranial vault [9]. The accuracy of the mathematical conformation of the generic mesh to extract the topography of the captured images has been previously evaluated and verified. The error of the conformation process was less than 1 mm [10]. In this study, the cranio-facial conformed meshes, consisted of 7461 vertices (7145 vertices for the facial mask and 316 vertices for the cranial mask). These were used for craniofacial analysis of the CZS and control cases. The conformation process was guided by manually digitizing 33 landmarks on the face and 6 landmarks on the vault, which initiated the elastic deformation of the mesh [7]. The following measurements of the cranial vault and face were carried out:Centroid size, it is the square root of the sum of squared distances from all the vertices of a mesh to the mathematical center of the mesh known as the centroid.The head circumference (HC), which is routinely measured from the most anterior point of the forehead to the most posterior point of the back of the head.The head height (HH), the coronal distance from the right ear to left ear across the highest point of the vault of the skull.The total facial height (TFH), is the distance from ophryon to gnathion.The chin height (CH), the distance between the gnathion and the mentolabial fold.

The measurements were repeated by the same assessor to evaluate the errors of the method.

### 2.3. Statistical Analysis of the Measurements

Student’s *t*-test was applied to compare the measurements between the study and the control groups. Pearson correlations between the measurements and ratios were calculated, with a significance level of 0.05. MATLAB 2024a (MathWorks^®^ MA 01760-2098 United States), with the statistics and machine learning toolbox, used for the statistical analysis. Receiver operating characteristic (ROC) curves were plotted to evaluate the CZS diagnostic performances of the ratios HH/HC and CH/HC. The ROC curve was generated by calculating and outlining the true positive rate (TPR) and false positive rate (FPR), at various thresholds. The Area Under the ROC Curve (AUC) allowed the comparison between the two models as well as the evaluation of the same model’s performance across different thresholds. AUC represents the degree of separability, the higher the AUC, the better the model. Partial Procrustes Alignment (PPA) was applied to measure the average head shapes of the CZS children and the controls [11]. PPA aligns the 3D head images to minimize the distances between their corresponding vertices in terms of the least square root of the total distances, without scaling individual 3D head images. The average head shapes of the CZS children and controls were also aligned using PPA to illustrate the differences in size and shape (Figure 1). Videos of morphing from the mean CZS shape to the mean control shape were generated and included in the Appendix A. 

## 3. Results

The study was successfully conducted on 35 Zika cases with an average age of 16.8 ± 2 months and 35 controls with an average age of 14.4 ± 5 months. The age range of the patients and controls was from 7 months to 24 months.

The errors of the repeated measurement of the HC, HH, CH, and TFH were less than 0.2 mm. The repeated landmarks’ digitization error was <0.5 mm.

The average head size of the CZS children was significantly smaller than that of the controls. The volume of the vault of the CZS cases was smaller than that of the control cases. The average control had a prominent forehead and a larger vault. In the CZS cases, the chin was less prominent, and the length of the mandible appeared shorter than that of the control cases. The width of the cranial vault from the right to the left sides and the total facial height were similar in the two groups. More details can be viewed in the videos included in the Appendix A. The visualized size differences were reflected in the results of the statistical analysis of the linear measurements and indices. Statistical analysis of the centroid sizes, which provided a measure of the geometric scale of the model, showed a significant difference (*p* = 0.0001) between the CZS cases and the controls (Table 1). There were significant differences in head circumference (HC *p* = 0.0000), head height (HH *p* = 0.0000), chin height (CH *p* = 0.0001), HH/HC (*p* = 0.0000), and CH/TFH (*p* = 0.0012) between the CZS cases and the controls. There was no significant difference in total facial height between the two groups (TFH, *p* = 0.5169).

There were significant correlations between the centroid size (r = 0.49, *p* = 0.0027), HC (r = 0.67, *p* = 0.0000), HH (r = 0.45, *p* = 0.0073), and TFH (r = 0.44, *p* = 0.0083) and the ages of the infants in the control group. There were no significant correlations between these measurements and the ages of the CZS cases. No significant correlations were detected between HH/HC or CH/TFH and the ages of either the control group or the CZS cases.

To find the optimal cut-off point for diagnosing CZS, Receiver Operating Characteristic (ROC) curves were plotted (Figure 2). The AUC of HH/HC was 0.88. Its optimal cut-off for the HH/HC was 0.49, with a sensitivity of 0.86 and a specificity of 0.74. The AUC of CH/TFH was 0.71. Its optimal cut-off for the CH/TFH was 0.25, with a sensitivity of 0.54 and a specificity of 0.85.

## 4. Discussion

Congenital Zika virus infection is one of the most challenging global infectious epidemics [12]. The WHO recommended and improved surveillance to better ascertain the incidence of ZKV transmission [13]. One of the characteristic features of CZS is severe microcephaly with a partially collapsed skull [2]. HC is a standard measurement that is routinely used to discriminate microcephaly from normal infants; however, research showed that it was not the best measurement that correlates with distinctive cranial morphology [7]. Franca et al. [14] reported that 20% of congenital Zika virus infections were associated with an HC that falls within the normal range, according to the international fetal and newborn WHO growth chart. It has been demonstrated that the HC measurement and the available norms lack sensitivity and specificity, in addition to the significant variation in HC among ethnic groups [13,15]. It is important for pediatricians to recognize the craniofacial phenotype of Zika cases to ensure appropriate, timely management and follow-up. It is crucial to find a new method to identify those 20% of “normal measurements” in a larger data set. The method described in this study is more sensitive than the routinely used measuring tape to assess head circumference.

Martini et al. [16] reported that an additional measurement of the height of the vault of the skull from the right to the left ear (HH) strengthens the ability of the model to predict the volume of the head for the monitoring of cranial growth. Hammond et al. [17] suggested that the 3D models provide dramatic visualizations of 3D face-shape variation with the potential of training physicians to recognize the key components of particular craniofacial syndromes. The 3D image system that we used in this study allowed the accurate, reliable, and fast capture of facial morphology without exposing the infants to harmful radiation. It enabled detailed morphometric analysis of the Zika microcephaly, which is crucial for diagnosing and understanding this particular anomaly. In this study, we made use of 3D imaging of the head to comprehensively evaluate the mean differences in the CZS cases in comparison with the control cases, which disclosed the characteristics of the face and cranial vaults of ZKV-infected cases. The visualized morphometric features have been confirmed with the results of the statistical analysis of the linear measurements and the assessed indices. The captured images allowed the visualization of the face and the skull in 360 degrees and facilitated the comparative analysis between the CZS and control cases. The 3D imaging of the head would inform clinicians about the level of brain abnormalities, as we have shown in our previous study (Ayoub et al. [7]. This would help to stratify the cases, according to the severity of the neural deficit, for a specialized management protocol.

Ayoub et al. [7] analyzed HH on the 3D models of CZS cases and proved that the inclusion of the additional measurement of HH has improved the accuracy of detecting ZKV-infected infants. Fonteles et al. [8] investigated the facial pattern of CZS cases; they reported that a CH/TFH of 0.229 showed 93.9% sensitivity and 80.6% specificity in diagnosing CZS. In this study, we analyzed the centroid size, HC, HH, CH, TFH, HH/HC, and CH/TFH. Apart from TFH, we demonstrated the significant differences in these measurements between the CZS children and controls. Correlations of linear measurements to the ages of the children in the control group were significant, but the indices of HH/HC and CH/TFH were not. Since HH/HC and CH/TFH were not correlated to the ages, it would be more efficient to use these in the surveillance of CZS. Only a single value is needed to classify CZS cases and healthy controls. Based on the ROC curves, we found that HH/HC had a better performance in separating CZS cases from healthy controls, with an AUC of 0.88, a sensitivity of 0.86, and a specificity of 0.74. On the other hand the CH/TFH had a poorer performance than what has been reported [8]. The study showed that HH is more sensitive as one of the diagnostic aids for microcephaly. We agree that the use of measuring tape for HH/HC is not ideal, yet it is readily available and routinely used in maternity units. It is possible to use hand-held 3D imaging devices to capture the morphology of the vault of the skull and measure HH more efficiently. This does not expose the infants to any harmful radiation and can be repeated as often as required. We hope the findings of this study inspire a paradigm shift in the diagnosis of craniofacial anomalies through the use of advanced 3D image capture and analysis technology.

## 5. Conclusions

There are significant differences in the size of the vault of the skull and the face between the CZS children and controls. The index of HH/HC has great potential for improving the sensitivity of the diagnosis of CZS cases. The HH measurement and the HH/HC index will aid in the screening and the diagnosis of microcephaly. The detection of this anomaly is pertinent in the diagnosis, grading, and management of other craniofacial syndromes. Hand-held 3D image capture devices should be routinely available in maternity units to facilitate comprehensive analysis of craniofacial morphology.

## Figures and Tables

**Figure 1 viruses-16-01406-f001:**
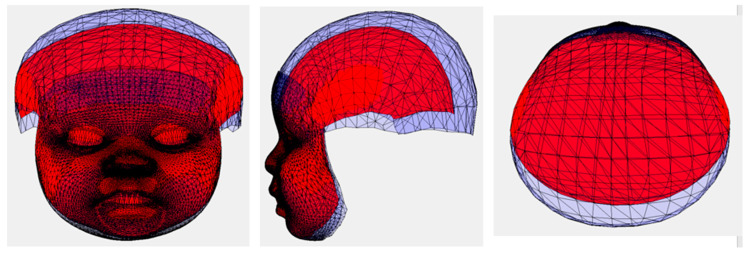
The differences between the average 3D models of CZS (red) and Control (blue).

**Figure 2 viruses-16-01406-f002:**
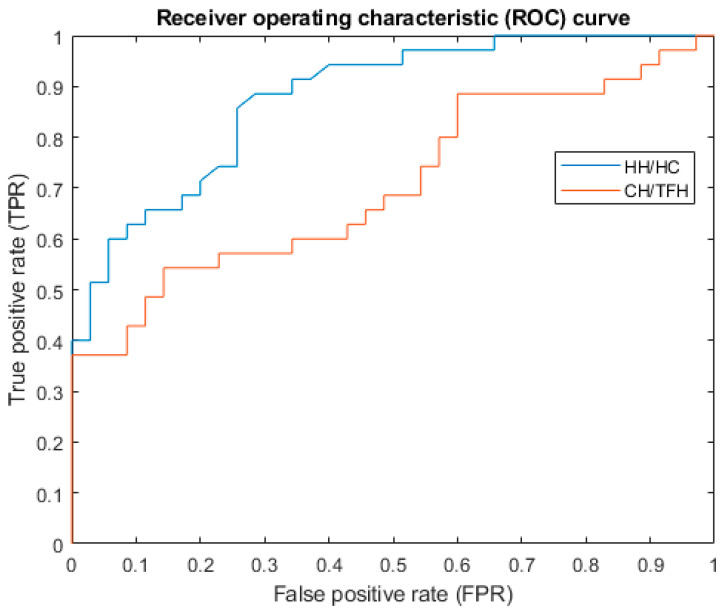
ROC curves of HH/HC and CH/TFH.

**Table 1 viruses-16-01406-t001:** Means and standard deviations of the measurements.

Measurements	CZS Group	Control Group	*p*-Value (Student’s *t*-Test with a Significant Level 0.05)
Centroid size	3471.4 mm ± 180.9	3641 mm ± 160.6	0.0001
HC	396.6 mm ± 27.1	468.3 mm ± 19.6	0.0000
HH	183.6 mm ± 20.2	240.0 mm ± 20.1	0.0000
TFH	83.5 mm ± 8.3	84.6 mm ± 6.2	0.5169
CH	20.1 mm ± 4.2	23.5 mm ± 2.6	0.0001
HH/HC	0.46 ± 0.03	0.51 ± 0.03	0.0000
CH/TFH	0.24 ± 0.05	0.28 ± 0.04	0.0013

## Data Availability

The data are available in the Appendix A.

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
