# Peer review of "3D Head Shape Feature Analysis of Zika-Infected Children"

_viruses, 2024, doi:10.3390/v16091406_

Round 1

Reviewer 1 Report

Comments and Suggestions for Authors

 This manuscript focused on identifying potential diagnostic measurements for microcephaly caused by congenital zika virus infection, through 3D image analysis.

The authors identified useful indexes with potential to be used as the gold-standard for the screening of CZS cases. This is a relevant contribution, since the current screening for CZS is based mainly in the measurement of the head circumference. As the cranial vault in CZS frequently has abnormal shapes, the HC might not be a sensitive measurement. 3D analysis is an affordable and easy technology and can be applied in different settings. The methodology is adequate and supports the findings and conclusions  

Author Response

Comment:  This manuscript focused on identifying potential diagnostic measurements for microcephaly caused by congenital zika virus infection, through 3D image analysis.

The authors identified useful indexes with potential to be used as the gold-standard for the screening of CZS cases. This is a relevant contribution, since the current screening for CZS is based mainly in the measurement of the head circumference. As the cranial vault in CZS frequently has abnormal shapes, the HC might not be a sensitive measurement. 3D analysis is an affordable and easy technology and can be applied in different settings. The methodology is adequate and supports the findings and conclusions .

Response: We agree with the comment. Thank you!

Reviewer 2 Report

Comments and Suggestions for Authors

Ju and colleagues performed a 3D head shape analysis of children with a history of congenital infection with Zika virus (ZKV) and compared the children to those without known exposure. The work is built on their previous efforts and confirms the clinical impression that the craniofacial phenotype associated with congenital Zika syndrome is most notable for loss of cranial column. This work could possibly improve screening methods and provide better insight into the mechanism producing the phenotype.

Comments

Abstract

Line 9. CZS is better described as a constellation of congenital anomalies rather than a malformation. Some of the brain anomalies, such as neuronal migration defects, can be characterized as a malformation but most anomalies more appropriately as disruptions.

Line 10-11. The sentence that ZKV infection could lead to microcephaly which impacts development relates to the more common situation in which primary microcephaly is associated with subsequent restricted brain growth. With congenital ZKV infection the sequence of events is somewhat reversed with a severe decrease in brain volume leading to microcephaly.   

Line 14. Assume these infants/children are not currently ZKV positive but were determined to have congenital ZKV infection shortly after birth. Please see comment under methods.

Lines 21-22. Please see comment under Discussion related to the abstract conclusion.

Introduction

Paragraph 2.

Please review this paragraph as there are misstatements and statements with unclear meaning.

Lines 31-32. The meaning of the first sentence about the need for initial screening for diagnosis of CZS with absence of serologic evidence is unclear. Do the authors mean that infants with craniofacial or neurologic abnormalities who test negative for Zika virus should be screened – and by what screening method?

Lines 32-33. The five features of CZS mentioned were rarely seen in other congenital infections OR were unique to congenital Zika infection.

Lines 37-40. The statement that some features of CZS are common to other congenital malformations seems incongruous with the subsequent statement that diagnosis was based on microcephaly (which is perhaps the most common dysmorphic finding in other congenital infections or genetic syndromes) and these statements aren’t supported in reference 3.

Paragraph 3.

Line 43. Please reference the statement about timing of HC measurement. At one point WHO was recommending HC on day 3, I believe, however, their own standards were developed from measurements obtained before 24 hours of age.

Lines 43-45. Please clarify the sentence – 20% normal HC may be within the normal range in 20% of children with congenital microcephaly – review of reference 4 does not help clarify.

Lines 45-47. Does the third sentence refer to reference 4, e.g., who felt the diagnostic criteria were nonspecific?

Lines 47-48. There are trade-offs between specificity and sensitivity in surveillance and it’s not clear that strengthening sensitivity should be the only goal.

Materials and Methods

Please describe the initial diagnostic assessment of case infants. Was the clinical diagnosis confirmed by laboratory – NAAT and/or Zika-related IgM?

Results

Line 130. Infants who have overlapping sutures due to collapse of the cranium have been found to have secondary craniosynostosis (Masoumy et al., FACE 2020;1(1):44-50). Could this have impacted study findings in cohort of case infants whose ages range from 7 to 24 months?

Discussion

Lines 171-172. Some infants with congenital ZKV infection and brain anomalies consistent with CZS have a HC at birth measuring in the normal range but develop microcephaly during the first year of life (van der Linden et al, MMWR. 2016 Dec 2;65(47):1343-1348). The microcephaly phenotype in these infants appears to be more consistent with that seen in most other types of congenital infections. It seems unlikely that these infants would be detected at birth by the methods described in this study.

Lines 208-209. How can this finding be translated into clinical practice for early screening? 3D analysis will not be available. Multiple studies have demonstrated the difficulties of obtaining an accurate HC – including the difficulty of obtaining simple non-stretchable measuring tapes. Apparently HH be measured with a measuring tape (reference 15) but are the same errors as in measuring HC expected?  And are there robust standards that could be used for HH or HH/HC measures in clinical settings?

Conclusions

Line 214-215. Although there is the potential to diagnose infants/children with CZS, other congenital infections such as CMV occasionally have been documented to cause a similar craniofacial phenotype and brain anomalies. The major difference in the phenotypes in these two congenital infections is the location of brain calcifications, i.e., subcortical versus periventricular).

Minor Comments

Please add the term “congenital” to Zika virus infection throughout the manuscript.

Line 10. Zika virus is most commonly abbreviated as ZKV.

Lines 11-12. There are two sentences connected by a comma.

Line 13. The measurement cut-off is 2 SD below the mean, not the norm.

Line 26. Please spell out World Health Organization when first used.

Lines 206, 210. Is the abbreviation TFG a typo?

Comments on the Quality of English Language

There are a few sentences needing clarification and those are noted in the review.  Otherwise, no issues with language.

Author Response

Ju and colleagues performed a 3D head shape analysis of children with a history of congenital infection with Zika virus (ZKV) and compared the children to those without known exposure. The work is built on their previous efforts and confirms the clinical impression that the craniofacial phenotype associated with congenital Zika syndrome is most notable for loss of cranial column. This work could possibly improve screening methods and provide better insight into the mechanism producing the phenotype.

Comments

Comment 1: Line 9. CZS is better described as a constellation of congenital anomalies rather than a malformation. Some of the brain anomalies, such as neuronal migration defects, can be characterized as a malformation but most anomalies more appropriately as disruptions.

Response 1: We agree with the comment and have changed to “a constellation of congenital anomalies” instead of “a malformation”.

Comment 2: Line 10-11. The sentence that ZKV infection could lead to microcephaly which impacts development relates to the more common situation in which primary microcephaly is associated with subsequent restricted brain growth. With congenital ZKV infection the sequence of events is somewhat reversed with a severe decrease in brain volume leading to microcephaly.  

Response 2: We agree with the comment and have changed to “The infection with ZV could lead to microcephaly of the foetus due to a severe decrease in brain volume and the reduced brain growth”.

Comment 3: Line 14. Assume these infants/children are not currently ZKV positive but were determined to have congenital ZKV infection shortly after birth. Please see comment under methods.

Response 3:  Infants of positive Zika-related IgM were recruited. We added the sentence in the methods section.

Comment 4: Lines 21-22. Please see comment under Discussion related to the abstract conclusion.

Response 4: We addressed the comments in discussion.

Comment 5: Please review this paragraph as there are misstatements and statements with unclear meaning. Lines 31-32. The meaning of the first sentence about the need for initial screening for diagnosis of CZS with absence of serologic evidence is unclear. Do the authors mean that infants with craniofacial or neurologic abnormalities who test negative for Zika virus should be screened – and by what screening method?

Response 5 We removed this sentence.

Comment 6: Lines 32-33. The five features of CZS mentioned were rarely seen in other congenital infections OR were unique to congenital Zika infection.

Response 6: We agree and changed “and” to “or”.

Comment 7: Lines 37-40. The statement that some features of CZS are common to other congenital malformations seems incongruous with the subsequent statement that diagnosis was based on microcephaly (which is perhaps the most common dysmorphic finding in other congenital infections or genetic syndromes) and these statements aren’t supported in reference 3.

Response 7: We agree and have rewritten the sentence and moved it to next paragraph.

Comment 8: Paragraph 3. Line 43. Please reference the statement about timing of HC measurement. At one point WHO was recommending HC on day 3, I believe, however, their own standards were developed from measurements obtained before 24 hours of age.

Response 8: The first sentence has been changed to “It is recommended that the head circumference should be measured within 24 hours of birth to identify congenital microcephaly [3]”.  Reference has been added - Sengasai C, Chokephaibulkit K, Plipat N, Wongsiridej P. Serial head circumference measurements should be used to classify congenital microcephaly. BMC Pediatr. 2023 Sep 27;23(1):490. doi: 10.1186/s12887-023-04315-4. PMID: 37759153; PMCID: PMC10523790.

Comment 9: Lines 43-45. Please clarify the sentence – 20% normal HC may be within the normal range in 20% of children with congenital microcephaly – review of reference 4 does not help clarify.

Response 9: We changed the sentence to “However, about 20% of infants with a normal HC have developed microcephaly at a later stage [5].”.

Comment 10: Lines 45-47. Does the third sentence refer to reference 4, e.g., who felt the diagnostic criteria were nonspecific?

Response 10: The sentence has been changed to “Therefore, the current measuring of HC is relatively unspecific and based on skull phenotype which is too broad to capture microcephaly of CZS”.

Comment 11: Lines 47-48. There are trade-offs between specificity and sensitivity in surveillance and it’s not clear that strengthening sensitivity should be the only goal.

Response 11: We agree and added “specificity”.

Comment 12: Please describe the initial diagnostic assessment of case infants. Was the clinical diagnosis confirmed by laboratory – NAAT and/or Zika-related IgM?

Response 12: A sentence was added “Infants of positive Zika-related IgM were recruited”.

Comment 13: Line 130. Infants who have overlapping sutures due to collapse of the cranium have been found to have secondary craniosynostosis (Masoumy et al., FACE 2020;1(1):44-50). Could this have impacted study findings in cohort of case infants whose ages range from 7 to 24 months?

Response 13:  Unlikely, the mechanism of microcephaly secondary to craniosynostosis is different from ZKV which impacts on brain development.

Comment 14: Lines 171-172. Some infants with congenital ZKV infection and brain anomalies consistent with CZS have a HC at birth measuring in the normal range but develop microcephaly during the first year of life (van der Linden et al, MMWR. 2016 Dec 2;65(47):1343-1348). The microcephaly phenotype in these infants appears to be more consistent with that seen in most other types of congenital infections. It seems unlikely that these infants would be detected at birth by the methods described in this study.

Response 14: We added “The method described in this study is more sensitive than the routinely used measuring tape to assess head circumference”.

Comment 15: Lines 208-209. How can this finding be translated into clinical practice for early screening? 3D analysis will not be available. Multiple studies have demonstrated the difficulties of obtaining an accurate HC – including the difficulty of obtaining simple non-stretchable measuring tapes. Apparently HH be measured with a measuring tape (reference 15) but are the same errors as in measuring HC expected?  And are there robust standards that could be used for HH or HH/HC measures in clinical settings?

Response 15: We added “The study showed that HH is more sensitive as one of the diagnostic aids of microcephaly. We agree that the use of measuring tape of HH/HC is not ideal, yet it is readily available and routinely used in maternity units. It is possible to use the hand-held 3D imaging devices to capture the morphology of the vault of the skull and measure HH. This does not expose the infants to any harmful radiation and can be repeated as frequent as required. We hope the findings of this study inspires the paradigm shift in the diagnosis of craniofacial anomalies using the technological advances 3D image capture and analysis”.

Comment 16: Line 214-215. Although there is the potential to diagnose infants/children with CZS, other congenital infections such as CMV occasionally have been documented to cause a similar craniofacial phenotype and brain anomalies. The major difference in the phenotypes in these two congenital infections is the location of brain calcifications, i.e., subcortical versus periventricular).

Response 16: We agree.

Minor Comments

Comments 17: Please add the term “congenital” to Zika virus infection throughout the manuscript.

Response 17:  We added the term “congenital”. Thank you.

Comment 18: Line 10. Zika virus is most commonly abbreviated as ZKV.

Response 18: Agree, thank you.

Comment 19: Lines 11-12. There are two sentences connected by a comma.

Response 19:   Corrected. Thank you.

Comment 20: Line 13. The measurement cut-off is 2 SD below the mean, not the norm.

Response 20: Agree, has been corrected.

Comment 21: Line 26. Please spell out World Health Organization when first used.

Response 21: Thank you, done.

Comment 22: Lines 206, 210. Is the abbreviation TFG a typo?

Response 22: Corrected to TFH.

Reviewer 3 Report

Comments and Suggestions for Authors

The infection with Zika virus (ZIKV) could result in fetal microcephaly, impacting brain development. This study aims to identify potential diagnostic measures for Zika Children. However, the study suffers from significant experimental design flaws: the authors should clearly outline the grouping, selection criteria of each group, and methods for determining congenital Zika infection. Moreover, Fonteles et al. have previously studied similar data, diminishing the novelty of this work.

Abstract

Line 9: The abbreviation for Zika virus should be corrected to ZIKV. Lines 19-20: The difference between HH/HC can only suggest a high likelihood of CZS (Congenital Zika Syndrome), not be used for its diagnosis. Microcephaly is only one symptom and is not exclusive to CZS.

Materials and Methods

1. Who constituted the control group? 2. How was Zika virus infection determined in children? How were congenital and acquired infections distinguished? Please provide clarification. 3. The study lacks children infected with Zika virus who did not develop CZS, limiting the conclusions drawn.

Results

1. What does P=0.0000 signify? Adding appropriate statistical symbols would enhance understanding of the statistical significance here. 2. Where are the data on children's ages and measurement indicators?

Discussion

Line 205: What are "linear measurements"? This term hasn't been mentioned earlier. Line 206: TFG?

Formatting

1. There are many unnecessary spaces throughout the manuscript; please remove them. 2. The English need to be polished.

Author Response

Comment 1: The infection with Zika virus (ZIKV) could result in foetal microcephaly, impacting brain development. This study aims to identify potential diagnostic measures for Zika Children. However, the study suffers from significant experimental design flaws: the authors should clearly outline the grouping, selection criteria of each group, and methods for determining congenital Zika infection. Moreover, Fonteles et al. have previously studied similar data, diminishing the novelty of this work.

Response 1: We have made changes accordingly. We added “Infants of positive Zika-related IgM were recruited”, stated “A control group of healthy infants, from the same geographic region, were included in the study for comparison”.  The investigation by Fonteles et al., was limited to 2D linear measurements of facial features in CZS and did not measure the 3D morphology of vault of the skull, which is a major limitation. We investigate the CZS cases in 3D and the method described in this study is more sensitive, provides accurate measurement of the vault of the skull rather than the Euclidean distance that was measured by Fonteles et al. Therefore, the novelty of this study is clear, and the clinic impact of the findings is remarkable.

Comment 2: Line 9: The abbreviation for Zika virus should be corrected to ZIKV.

Response 2: We have changed “ZV” to “ZKV”.

Comment 3: Lines 19-20: The difference between HH/HC can only suggest a high likelihood of CZS (Congenital Zika Syndrome), not be used for its diagnosis. Microcephaly is only one symptom and is not exclusive to CZS.

Response 3: We agree. The diagnosis is multidisciplinary, the measurement of HH/HC is for the early screening of CZS cases to replace the routinely used head circumference measurement.

Comment 4: Who constituted the control group?

Response 4: We stated “A control group of healthy infants, from the same geographic region, were included in the study for comparison” in line 76-77.

Comment 5: How was Zika virus infection determined in children?

Response 5: We added “Infants of positive Zika-related IgM were recruited”.

Comment 6: How were congenital and acquired infections distinguished? Please provide clarification.

Response 6: We focused on the Congenital Zika Virus infected cases in this study.

Comment 7: The study lacks children infected with Zika virus who did not develop CZS, limiting the conclusions drawn.

Response 7: The conclusion of this study was based on the cohort of infants who proved infected with ZKV and developed microcephaly in addition to other neurological symptoms.

Comment 8:  What does P=0.0000 signify? Adding appropriate statistical symbols would enhance understanding of the statistical significance here.

Response 8: Table 1 has been changed by adding “(student t-test with a significant level 0.05)” next to “p-value”.

Response 9: Where are the data on children's ages and measurement indicators?

Response 9: Data were uploaded in the file of “zika_control measurements.xlsx” of the supplementary.

Comment 10: Line 205: What are "linear measurements"? This term hasn't been mentioned earlier. Line 206: TFG?

Response 10: Linear measurement is the measurement of the distance between the two points or objects. TFG was corrected to TFH.

Comment 11: There are many unnecessary spaces throughout the manuscript; please remove them.

Response 11: We tried to remove unnecessary spaces manually.

Comment 12: The English need to be polished.

Response 12: We asked a native English speaker to review the English.